# A Role for NF-κB in Organ Specific Cancer and Cancer Stem Cells

**DOI:** 10.3390/cancers11050655

**Published:** 2019-05-11

**Authors:** Christian Kaltschmidt, Constanze Banz-Jansen, Tahar Benhidjeb, Morris Beshay, Christine Förster, Johannes Greiner, Eckard Hamelmann, Norbert Jorch, Fritz Mertzlufft, Jesco Pfitzenmaier, Matthias Simon, Jan Schulte am Esch, Thomas Vordemvenne, Dirk Wähnert, Florian Weissinger, Ludwig Wilkens, Barbara Kaltschmidt

**Affiliations:** 1Department of Cell Biology, Bielefeld University, Universitätsstrasse 25, 33615 Bielefeld, Germany; johannes.greiner@uni-bielefeld.de (J.G.); barbara.kaltschmidt@uni-bielefeld.de (B.K.); 2Department of Gynecology and Obstetrics, and Perinatal Center, Protestant Hospital of Bethel Foundation, Burgsteig 13, 33617 Bielefeld, Germany; Constanze.Banz-Jansen@evkb.de; 3Department of General and Visceral Surgery, Protestant Hospital of Bethel Foundation, Schildescher Strasse 99, 33611 Bielefeld, Germany; tahar.benhidjeb@evkb.de (T.B.); jan.schulteamesch@evkb.de (J.S.a.E.); 4Department for Thoracic Surgery and Pneumology, Protestant Hospital of Bethel Foundation, Burgsteig 13, 33617 Bielefeld, Germany; Morris.Beshay@evkb.de; 5Institute of Pathology, KRH Hospital Nordstadt, Haltenhoffstrasse 41, affiliated with the Protestant Hospital of Bethel Foundation, 30167 Hannover, Germany; Christine.Foerster@evkb.de (C.F.); ludwig.wilkens@krh.eu (L.W.); 6Children’s Center, Protestant Hospital of Bethel Foundation, 33617 Bielefeld, Germany; Eckard.Hamelmann@evkb.de (E.H.); norbert.jorch@evkb.de (N.J.); 7Scientific Director, Protestant Hospital of Bethel Foundation, Maraweg 21, 33617 Bielefeld, Germany; Fritz.Mertzlufft@evkb.de; 8Department of Urology and Center for Computer-assisted and Robotic Urology, Protestant Hospital of Bethel Foundation, Burgsteig 13, 33617 Bielefeld, Germany; Jesco.Pfitzenmaier@evkb.de; 9Department of Neurosurgery and Epilepsy Surgery, Protestant Hospital of Bethel Foundation, Burgsteig 13, 33617 Bielefeld, Germany; Matthias.Simon@evkb.de; 10Department of Orthopedics, Trauma Surgery, and Trauma Center, Protestant Hospital of Bethel Foundation, Burgsteig 13, 33617 Bielefeld, Germany; Thomas.Vordemvenne@evkb.de (T.V.); Dirk.Waehnert@evkb.de (D.W.); 11Department of Hematology, Oncology, Internal Medicine, Bone Marrow and Stem Cell Transplantation, Palliative Medicine, and Tumor Center, Protestant Hospital of Bethel Foundation, Burgsteig 13, 33617 Bielefeld, Germany; Florian.Weissinger@evkb.de; 12Molecular Neurobiology, Bielefeld University, Universitätsstrasse 25, 33615 Bielefeld, Germany

**Keywords:** cancer stem cells, NF-κB, glioblastoma multiforme, pediatric cancer, ovarian cancer, multiple myeloma, lung cancer, colon cancer, prostate cancer, bone cancer

## Abstract

Cancer stem cells (CSCs) account for tumor initiation, invasiveness, metastasis, and recurrence in a broad range of human cancers. Although being a key player in cancer development and progression by stimulating proliferation and metastasis and preventing apoptosis, the role of the transcription factor NF-κB in cancer stem cells is still underestimated. In the present review, we will evaluate the role of NF-κB in CSCs of glioblastoma multiforme, ovarian cancer, multiple myeloma, lung cancer, colon cancer, prostate cancer, as well as cancer of the bone. Next to summarizing current knowledge regarding the presence and contribution of CSCs to the respective types of cancer, we will emphasize NF-κB-mediated signaling pathways directly involved in maintaining characteristics of cancer stem cells associated to tumor progression. Here, we will also focus on the status of NF-κB-activity predominantly in CSC populations and the tumor mass. Genetic alterations leading to NF-κB activity in glioblastoma, ependymoma, and multiple myeloma will be discussed.

## 1. Introduction

Cancer stem cells, also called neoplastic stem cells or cancer initiating cells, were discovered by transplantation in immunocompromised mice. Only a small fraction of all dissociated cells was propagated in the nude mouse model (1/250,000) [1]. Since one cell with markers for stem cells such as CD34 for leukemia or CD133 for solid cancers could initiate cancer growth, the concept of cancer stem cells (CSC) was born. Characteristics of CSCs are self-renewal, differentiation in other more mature cell types, presumable from different germ layers, and tumor initiation in suitable mouse model. In vitro propagation as spheres, dye exclusion and resistance to chemotherapeutics, and lack of MHC class I expression can be used for characterization [2,3,4].

Cancer stem cells manifest the capacity of self-renewal, DNA repair, persisting in the G1 or G0 cell cycle phases as inactive dormant cells, and asymmetric cell division. Interestingly, especially asymmetric cell division is discussed for being a hallmark of CSCs [5,6]. For instance, Takeda and colleagues recently reported 90% of Sox2-positive colon cancer stem cells to undergo asymmetric cell division. In this line, breast cancer stem cells express the receptor Notch, which could be stimulated by NF-κB-mediated expression of its ligand JAG1 on non-cancer stem cells. Thus, proliferation of CSCs can be triggered by an NF-κB-dependent mechanism [7]. As a further major hallmark, CSCs do not undergo apoptosis and they manifest overexpression of ABC genes, which is linked to their resistance to cytostatic drugs. Control of their self-replacement is associated in principle with numerous signaling pathways, including Notch, Sonic hedgehog (Shh), and wingless-type (Wnt). Cancer stem cells can be identified and isolated due to their specific markers, such as CD44, CD133 (prominin-1, see also Figure 3B), CD117 (c-Kit), ALDH1 (aldehyde dehydrogenase), and OCT3/4 (POU5F1), the transcription factor of the POU (Pit-Oct-Unc) family. In addition to these commonly accepted marker panels for CSC identification and isolation, increasing evidences suggest intracellular signaling pathways mediated by the transcription factor named “nuclear factor kappa-light-chain enhancer of activated B-cells” (NF-κB) to be of particular importance for CSC characteristics and functionality.

NF-κB is ubiquitously expressed and mediates a broad range of cellular processes ranging from apoptosis, cell growth, inflammation, memory, and learning to immunity [8,9]. The NF-κB family is characterized by a conserved n-terminal REL homology domain (RHD) being crucial for DNA-binding and dimerization of NF-κB family members. These family members particularly include the five subunits of NF-κB, namely RELA (p65), RELB, c-REL, p50 and p52, and the NF-κB. The NF-κB subunits RELA, RELB, and c-REL additionally comprise a C-terminal transactivation domain (TAD) [10]. As schematically depicted in Figure 1, inhibitors of κB (IκBs) mask the NLS (nuclear localization signal within the RHD) of NF-κB p50/p65 dimers, thereby preventing their nuclear translocation. Binding of ligands to their respective receptors (such as CD40) results in phosphorylation of the IκB kinase (IKK) complex (IKKγ/IKKα/IKKβ) in a C-IAP-, TRAF2/3-, and NIK (NF-κB-inducing kinase)-dependent manner. Phosphorylated IKKs in turn phosphorylate IκBα resulting in its proteasome-mediated degradation and demasking of the NLS within the p50/p65 NF-κB dimer. The NF-κB dimer is subsequently translocated into the nucleus and binds to specific target sites, thus enabling target gene expression [9,10]. Next to this canonical NF-κB signaling cascade, non-canonical NF-κB signaling is mediated by phosphorylation of IKKs via NIK, in turn leading to phosphorylation of p100 and its proteasomal processing to p52 [11] (see also Figure 1 for overview). Subsequent nuclear translocation of the p52/RELB NF-ĸB dimer is followed by binding to selective ĸB sites and activation of specific target genes. Various aspects of cancerogenesis and cancer progression are directly associated to deregulation of canonical and non-canonical NF-ĸB signaling pathways due to its various cellular functions and target genes [12,13,14,15,16]. In particular, canonical NF-ĸB signaling mediates vital tumor-promoting mechanisms like stimulation of cell proliferation and prevention of apoptosis, epithelial-to-mesenchymal transition (EMT), angiogenesis, invasiveness, as well as metastasis. As a driver of such crucial mechanisms inducing and propagating tumor growth, NF-κB was shown to be constitutively active in a broad range of cancers from various organs. Constitutive activity of NF-κB subunits in turn arises from a prolonged chronic inflammatory microenvironment or by various oncogenic mutations [12,17,18]. In chronic inflammation, accumulation of proinflammatory cytokines caused by increased activity of NF-κB was shown to promote a protumorigenic microenvironment in colon cancer [19]. However, NF-κB has also been described to have anti-inflammatory roles with direct effects on tumor formation and therapy resistance (reviewed in [20]). For instance, overexpression of the NF-κB p50 homodimer in tumor-associated M1 macrophages was reported to inhibit inflammatory and antitumor responses in murine fibrosarcoma. Defective responsiveness of tumor-associated macrophages to M1 activation signals in human ovarian carcinoma was also associated with activation of the NF-κB p50 homodimer [21], suggesting a context-dependent role of NF-κB-activity at least in cancers related to chronic inflammation.

In line with its essential role in cancer development and progression, increasing evidence suggests NF-κB to be pivotal in generation of CSCs and maintaining their functionality. With regards to maintenance of self-renewal of CSCs, Iliopoulos and coworkers discovered an epigenetic switch in mammospheres induced by overexpression of the SRC oncogene as a model of CSCs development in breast cancer [24]. In particular, a short period (24 h) of induction by SRC was sufficient to change gene expression in a manner that NF-κB and STAT3 signaling became dominant (see Figure 2). Furthermore, expression of transcription factor LIN28 was induced, which is a repressor of LET7 microRNA, thus releasing the repression of IL6 protein expression. Recent findings elaborated these observations to a more general model for the regulation of an inflammatory regulatory network in many human cancers (see Figure 2). This network involves three major inflammatory pathways: NF-kB, which can be activated by members of the TNF superfamily (TNFSF) and IL1, the transcription factor AP1 (JUN/FOS), and STAT3. All three transcription factors are activated by transmembrane receptors, which impinge on kinases (IKKs for NF-KB, JNK for AP1, and JAK for STAT3, see Figure 2 and [25]).

In line with the depicted feed-back loop of the repression of miRNA LET7 by embryonic transcription factor LIN28B in cancer stem cells (Figure 2), Markopoulos and coworkers recently reviewed more than 150 miRNAs acting on NF-κB in cancer [26]. Accordingly, we could previously show that NF-κB RelA protein is repressed by microRNAs in embryonic stem cells and that this repression is released during neuronal differentiation [27,28].

Among the broad range of NF-κB target genes and associated pathways, epithelial-to-mesenchymal transition (EMT) is one of the most prominent ones in regulating CSC-functionality [29,30]. Here, NF-κB regulates transcription of SNAIL, Slug, ZEB1, ZEB2, and Twist, which repress the epithelial phenotype and are directly associated to cancer invasiveness and aggressiveness as well as poor patient survival [30]. Regarding the induction of mesenchymal markers, NF-κB also regulates expression of Matrix metalloprotease (MMP) 2 and MMP9, as well as Vimentin. Particularly Vimentin-positive CSCs seem to represent the end stage progression of EMT, concomitant with a completely dedifferentiated state, which is highly proliferative and invasive [30]. In line with these observations, EMT was shown to generate cells with properties of stem cells from human mammary epithelial cells. Stem-like cells showed increased expression of stem cell markers, sphere formation ability, and more efficient capability for tumor formation [31]. Accordingly, blocking of NF-kappaB activity was reported to prevent EMT and metastatic potential of Ras-transformed mammary epithelial cells, likewise suggesting NF-κB activity to be critical for the cooperation of Ras- and TGF-beta-dependent signaling pathways [32].

Here, we limited ourselves to the KEGG pathway “NF-kappa B signaling pathway—Homo sapiens (human)” as depicted in [33,34]. Even here, we simplified the canonical and non-canonical pathways for the sake of eligibility. It might be clear that NF-κB is cross-coupled to several other pathways. For example, there is a negative regulation of NF-κB activity by glucocorticoid receptor signaling [35]. Another developmentally important pathway involving secreted protein ligands of the Wnt (wingless family) shows both negative and positive action on the NF-κB pathway (as reviewed in [36]). Wnt/β-catenin pathway components could regulate responses via the interaction with the NF-κB pathway. In turn, NF-κB could influence the activity of Wnt/β-catenin signaling pathway. Specifically, overexpression of beta catenin in breast and colon cancer could inhibit NF-κB activity by physical interaction, but there are reports on positive regulation [37]. Furthermore, in transformed mammary epithelial cells, TGF-β and Wnt signaling synergize to induce activation of the epithelial–mesenchymal transition (EMT) program, and function in an autocrine fashion to maintain the resulting stem-cell state [38]. The complex cross-talk between signaling pathways of Wnt, Notch, Hippo, Hedgehog (Hh), mitogen-activated protein (MAP), kinase, phosphoinositide 3-kinase (PI3K)-Akt, and the nuclear NF-κB has been covered in a recent review [39].

Emphasizing the pivotal roles of NF-κB in CSCs, the present review will focus on discussing commonly known as well as recently described CSC populations in a broad range of organ specific cancers (see Figure 3A for overview), starting with glioblastoma multiforme as the most frequent cancer type in brain. We focus on the contribution of distinct CSC-populations to cancer progression, thereby emphasizing the role of NF-κB as a crucial mediator for biology and function of CSCs.

## 2. Cancer Stem Cells in Glioblastoma Multiforme

Glioblastoma multiforme (GBM) is the most common primary brain cancer and particularly characterized by a notable cellular heterogeneity and aggressiveness as well as extensive invasiveness into neighboring brain tissue. The greatest problem is its inevitable recurrence. GBM patients have an average survival of less than two years [40,41,42]. In selected cases with a molecularly characterized tumor subtype, a combination therapy with lomustine and temozolomide may increase survival time to four years [43].

GBM contain a variety of poorly differentiated neural cells. Three major cell types are mostly recognized within GBM. This includes a population of relatively rare cancer stem cells, which are able to self-renew and repopulate the cancer. Next to cancer stem cells, quiescent cells, having exited the cell cycle, are likewise present in GBM. These quiescent GBM cells are characterized by a slow cellular turnover and no persistent self-renewal capacity. Both GBM CSCs and quiescent cells have the capability to give rise to the third pool of GBM cell types, the differentiated GBM cells. Differentiated GBM cells, which also include different stages of incomplete differentiation and respective progeny without tumor-propagating capacity, are the most prevalent cell pool in GBM and show mitotic activity. Stem-like cells can be found in the tumor infiltration zone (i.e., beyond the margins of the surgical resection), and therefore contribute prominently to tumor recurrence [44]. The differentiated GBM cell pool is considered as the main contributor to GBM tumor mass development [45,46,47,48,49,50].

Activation of NF-κB in glioblastoma was subject of a recent review [51]. Unregulated constitutive NF-κB activity is commonly observed in GBM and was proposed to be caused by several mechanisms. For instance, receptor tyrosine kinases like epidermal growth factor receptor (EGFR) or platelet derived growth factor receptor (PDGFR)are often constitutively activated in GBM. Involving both protein kinase B/AKT (AKT)-dependent and -independent pathways, activation of EGFR and PDGFR have been linked to NF-κB activation. EGFR gene amplification and overexpression of a constitutively active variant are considered molecular hallmarks of GBM, and have been explored as potential therapeutic targets [52]. In addition, there is a cellular crosstalk mediated by IL-6 and NF-κB, which leads to attenuation of EGFR tyrosine kinase inhibitors (TKEs) [53]. Targeting the NF-κB target gene IL-6 was also demonstrated to directly reduce survival of CSCs in GBM concomitant with decreased tumor growth [54]. Pyruvate kinase M2 (PKM2), a member of the enzyme family regulating the rate-limiting step of glycolysis, is overexpressed in numerous cancers and could be regulated in glioblastoma by NF-kB [51,55]. In addition, A20 (TNFAIP3), an NF-kB target gene and regulator of cell survival, was reported to be overexpressed in CSCs from GBM with elevated levels of A20 in GSCs contributing to apoptotic resistance [56]. Interestingly, a recent genetic analysis showed deletion of the IκBα (NFKBIA) gene in up to 24% of glioblastoma patients [57].

Happold and coworkers analyzed glioblastoma cancer stem cells and described NF-κB as a positive regulator of MGMT (O6-methylguanine DNA methyltransferase) mediated by NF-κB p65 [58]. Of note, MGMT promoter hypermethylation has proven an important predictive biomarker for benefit from alkylating chemotherapy as well as powerful prognostic factor [59,60]. This has resulted in attempts at therapeutic stratification of glioblastoma based on MGMT methylation status [43,61]. We could recently show that rat neurosphere cultures could be transformed by forced culture to a growth factor-independent characterized by constitutively activated NF-κB and production of VEGF [62]. In line with these findings, angiogenesis mediated by VEGF could be also regulated by NF-kB in glioblastoma [63]. Here, glioblastoma stem cells were recently shown to secrete VEGF-A in extracellular vesicles [64]. VEGF receptor inhibition as a therapeutic concept for glioblastoma has attracted much attention in recent years. The VEGF inhibitor bevacizumab has won FDA approval and is in widespread use in the US. Others are more critical, and several large prospective trials have failed to convincingly demonstrate a survival benefit both in primary as well as patients with recurrent glioblastoma. Many feel that bevacizumab reduces peritumoral edema, and has only very limited tumoricidal activity, at the least [61,65].

CSCs from glioblastoma seem to be highly resistant to the treatment with Smac mimetics, which are small molecule drugs inhibiting members of IAP (inhibitor of apoptosis) proteins. In CSCs from glioblastomas, Smac mimetics trigger an adaptive response with increased expression of TNFα and sustained activation of NF-κB and STA3 signaling [66]. The cross-talk between NF-κB and STAT3 drives tumor progression and promotes stemness characteristics of cancer in multiple malignancies including gliomas [67]. Furthermore, glioma stem cells secrete Sema3C and coordinately express Plexin A2/D1 receptors to activate Rac1/NF-κB signaling in an autocrine/paracrine loop to promote their own survival [68]). Another source of NF-κB activation in GBM involves the TGF-β/TAK1 signaling axis, which could activate both canonical and non-canonical NF-κB signaling [69]. Proneural (PN) and mesenchymal (MES) glioma stem cells (GSCs) represent two mutually exclusive and biologically distinct GSC subtypes. Recently, it was shown that mixed lineage kinase 4 (MLK4) is overexpressed in MES but not proneural PNGSCs. Silencing MLK4 suppressed self-renewal, motility, tumorigenesis, and radio-resistance of MES. MLK4 bound and phosphorylated the NF-κB regulator IKKα, leading to activation of NF-κB signaling in MES [70].

A novel NF-κB target gene is the telomerase reverse transcriptase (TERT). TERT can also activate NF-κB p65 (RelA), facilitating a feed forward loop (as reviewed in [14]), which enables increased binding to recently described TERT promotor polymorphism. TERT promoter mutations are thought to characterize primary as opposed to secondary GBM (i.e., GBM developing from a preexistent lower-grade glioma), and may play a role as a prognostic factor [71,72]. In line with these findings, TERT function was also shown to induce CSC characteristics in glioma cells via EGFR expression [73]. In summary, IKK/NF-κB signaling directly contributes to glioblastoma stem cell maintenance, as recently demonstrated by Rinkenbaugh and coworkers [69].

## 3. Cancer Stem Cells in Pediatric Cancer

Only around 0.5% of all newly diagnosed cases of cancer in Germany concern children or adolescents. This means that the overall incidence for cancer in this age group is around 14 of 100,000. Therefore, cancer in children may be appraised as a rare disease. Still, cancer is the second most frequent cause of death in patients beyond the first year of life after accidents. Since 1980, in Germany, all patients less than 18 years with cancer are enrolled in a specific register, so that we have detailed knowledge about entities, mortality rates, and treatment efficacy. Overall incidences in the last decades are not growing, although the number of patients with brain tumors has increased, most likely due to a better diagnosis and reporting system. By far the most frequent types of cancer in childhood contain lymphoblastic leukemia (nearly 30%), followed by astrocytoma (10%), neuroblastoma (8%), nephroblastoma and Non-Hodgkin-Lymphoma (6% each), non-lymphoblastic leukemia, and neuro-endocrine tumors (4% each) [74].

With 16,050 deaths per year, brain tumors were reported to be the leading cause of cancer death in children [75]. Here, glioblastoma multiforme (GBM; World Health Organization grade IV astrocytoma) is ranging among the most malignant and aggressive forms of brain tumors [76]. Opposite to the overall better survival rates of children with cancer, GBM survival rates have remained low for the last 50 years, with a median survival time of 12–18 months [77]. This explains why there is a big unmet need for more effective and individualized therapies that should be based upon the specific pathophysiology of the tumors and their microenvironment [78].

In pediatric tumors, acute myeloid leukemia stem cells were the first CSCs being identified [79] via expression of the hematopoietic stem cell marker CD34, while lacking the lymphocyte differentiation marker CD38 [80]. This initial report was followed by the observation of CSCs in a broad range of pediatric brain tumors, including medulloblastomas [3] and high-grade gliomas (HGGs) [81]. The technical proof for CSCs in pediatric brain tumors was similar as in adults: CSCs were isolated from bulk tumors using stem cell markers, then demonstrated their capacity for self-renewal, differentiation, and recapitulation of the original tumor [82].

CSCs in pediatric brain tumors express similar CSC markers as in adults (including CD133, SOX2, musashi-1, BMI1), but also show elevated expression of maternal embryonic leucine zipper kinase and phosphoserine phosphatase [83]. Moreover, epigenetic regulation plays a bigger role in pediatric than in adult brain tumors, since developing a brain tumor within the first two years of life is more likely to be the result of cancerogenic transformation caused by genetic and/or an epigenetic variation. On the contrary, an adult brain is rather long-term exposed to mutagenic transformation events occurring in oncogenes or tumor suppressor genes [84]. This diversity between childhood and adult brain tumors highlight that the CSC populations in these tumors most likely represent distinct targets for personalized therapeutic intervention. A role for NF-κB RELA in pediatric neuro-oncology is manifested by the RELA fusions in 70% of supratentorial ependymoma [85,86]. But only C11orf95–RELA fusions led to the development of ependymoma in a mouse model, whereas overexpression of C11orf95, or RELA or other fusions did not drive tumorigenesis in this model [87]. In particular, C11orf95–RELA fusion proteins were shown to activate NF-κB target genes in neural stem cells, leading to their transformation and initiation of tumor growth [85]. In Wistar rats, NF-κB is developmentally regulated during cerebellum development with high constitutive activity at p5 in neuronal stem cells, which drops dramatically during later life, suggesting a potential correlation with the effect of RELA fusion [88].

## 4. Ovarian Cancer and Cancer Stem Cells

Each year, 240,000 women are diagnosed with ovarian cancer worldwide and 150,000 die of it. Further, 47% of all deaths from cancer of the female genital tract occur in women with ovarian cancer, although only 23% of gynecologic cancers are ovarian in origin. Epithelial ovarian cancer is responsible for 5% of all cancer-related deaths in women [89]. The incidence rate is highest in high-income countries and increases with age. The largest number of patients with epithelial ovarian cancer in high-income countries is found at 60–64 years of age; in low-income countries, the median age is about one decade earlier [90]. Risk factors include reproductive factors. Women who have never given birth are twice as likely to develop ovarian cancer. Early menopause and the use of oral contraceptives are associated with a lower risk for the disease [91].

Notably, germline mutations in BRCA1/2 were observed in at least 15% of women with high grade non-mucinous ovarian cancers, despite that 40% of these women did not have a family history of breast or ovarian cancer [92]. The risk for developing ovarian, tubal, or peritoneal cancer for women with BRCA1 deleterious mutations is around 20–50% and 10–20% with a BRCA2 mutation. These cancers occur at an earlier age with a median age of diagnosis in the mid-40s [93]. Notably, BRCA1 was shown to associate constitutively with NF-κB p65, while p50 associated only with BRCA1 upon DNA damage treatment. Harte and coworkers thus described a functional interdependence between BRCA1 and NF-κB, which is required for BRCA1-mediated resistance to DNA damage [94]. In addition, NF-κB was demonstrated to directly regulate DNA double-strand break repair in a BRCA2-dependent manner as well as via interaction with CtIP-BRCA1 complexes and promotion of BRCA1 stabilization [95]. There are no effective screening methods that reduce the mortality of ovarian cancer [96]. The prognosis of ovarian cancer is dependent on stage of the cancer, histologic type and grade, and maximum diameter of residual disease after cytoreductive surgery [97].

Surgery includes the staging laparotomy and debulking surgery at the same time: All peritoneal surfaces are evaluated, peritoneal fluid or ascites is retrieved, (at least) infracolic omentectomy and lymphadenectomy of the pelvic and para-aortal lymph nodes, total hysterectomy, and bilateral salpingo-oophorectomy is performed. For mucinous tumors, appendectomy is mandatory. Optimal cytoreduction may necessitate bowel resection and resection of other organs or structures [98]. In selected patients with cytologically proven advanced disease, neoadjuvant chemotherapy may be given initially, followed by interval debulking surgery, and additional chemotherapy [99]. Patients who have had primary cytoreductive surgery for advanced ovarian cancer should receive platinum-based combination chemotherapy following surgery [100]. Bevacizumab, an antibody against VEGF-A, may be added to this regimen [101].

After cisplatin treatment, the dormant cells of human ovarian cancer displayed an augmented expression of cells with Oct-4, nestin, CD-117, and CD44 markers, proving the resistance of this stem cell fraction to cisplatin [102]. Thus, additional treatments with drugs more specific to CSC such as Salinomycin and/or Metformin might be considered [103]. In this line, it could be shown that Salinomycin inhibits NF-κB and induces apoptosis in cisplatin resistant ovarian cancer cells [104]. Furthermore, safety of treatment with allogenic in vitro cultured NK cells delivered into the intraperitoneal space by laparoscopy was studied [105]. It was shown, using CSC from ovarian cancers by magnetic sorting (anti CD133), that NF-κB-mediated migration is activated by an autocrinous loop of CCL5 to RelA [22] (Figure 1). Interestingly, CSCs can differentiate into endothelial cells under the influence of CCL5 [106]. Both RelA and RelB support ovarian cancer growth in a mouse model [107]. Blocking of classical NF-κB pathway results in reduction of CD44^+^ CSCs in ovarian cancer [108].

## 5. The Role of Cancer Stem Cells in Multiple Myeloma

Multiple myeloma is a monoclonal B-Cell neoplasia, characterized by proliferation of plasma cells and production of complete or incomplete antibodies (see Figure 1). The proliferation of plasma cells might lead to osteolytic lesions, renal insufficiency, hypercalcemia, and hematopoietic insufficiency. The incidence in Germany is about 6500 newly diagnosed diseases per year (http://gekid.de/download/1112/). Precursor stages of multiple myeloma could be MGUS (monoclonal gammopathy of unclear significancy) and smoldering myeloma. In contrast to a merely specific disease, multiple myeloma depicts genetic heterogeneity. In about 40% of cases, different kinds of trisomy are found. A chromosome 14 translocation is frequently observed, particularly containing the immunoglobulin heavy-chain locus like t(11;14), t(4;14), t(14;16), t(6;14), t(14;20). These chromosomal aberrations are found in MGUS also [109]. Furthermore, it can be discussed, that secondary aberrations like RAS mutations and translocations involving MYK could influence the course of disease and prognosis of myeloma [110]. There are different hypotheses for the genesis of multiple myeloma. The classic view on the evolution of multiple myeloma focuses on mature plasma cells in the bone marrow, where oncogenic mutations transformations took place. In summary, multiple myeloma was mostly described as a tumor arisen from transformed plasma cells [23]. Another emerging concept focuses the myeloma stem cell (MMSC), which is able to self-renew and differentiate into myeloma plasma cells. The MMSC is therefore categorized as a progenitor cell of the plasma cell. These cells are summarized as cancer initiating cells or cancer stem cells. A landmark study by Bonnet and Dick described a hierarchy of cells within human acute myeloid leukemia [80]. By transfer in immunocompromised mice, they identified a leukemia stem cell that was almost exclusively CD34^+^; CD38^−^. Most interestingly, in multiple myeloma patients, peripheral blood lymphocytes express high levels of CD34, with levels up to 37% in comparison to 5% in normal controls. Up to 98% of these CD34^+^ cells are positive to CD19 [111]. Taken together these data suggest the presence of malignant MMSC which could give rise to metastasis. Based on these findings, a multination consortium of clinicians suggested a selected sampling of biomaterial of different clinical stages and disease spectrum [112]. We plan to analyze the contribution of NF-κB to MMSC. In this line, most NF-kB activating mutations were attributed to mutations acquired by a stromal independent multiple myeloma cell (see [23] for discussion). Notably, canonical NF-κB signaling was already observed to be required for persistence functionality of mature B cells [113] and is a hallmark of various B cell lymphomas [114]. NF-κB cRel is further known as a key regulator of B-cell proliferation and is frequently amplified in B-cell lymphoma [115], but was also described to have a tumor suppressor role in lymphoma development [116]. In Figure 1, we summarize hypotheses for the generation of multiple myeloma and the activation of NF-κB.

## 6. Lung Cancer and Cancer Stem Cells

Lung cancer is the second most common malignancy in males after prostatic cancer and, surprisingly, it is now the second most common malignancy in females after breast cancer. According to its histological differentiation, it is classified into two main categories; small-cell lung cancer (SCLC) (less frequent form of the disease with an incidence of about 20% of all lung cancers) and the non-small-cell lung cancer (NSCLC) category (most frequent form with an incidence of about 80%). Treatment depends on the stage of the disease. In early stages (I and II), surgery, if possible, is the “gold standard” treatment. In higher stages (IIIA and IIIB), neoadjuvant (induction, preoperative) chemotherapy/radio-chemotherapy is an established way for better outcomes. In case of advanced stages especially with multiple metastases, palliative chemotherapy is the main strategy. In the last few years, target therapy or immune therapy has been gaining popularity but is still in its primitive age. In many cases, adjuvant therapy is needed after surgery. Nevertheless, the overall prognosis of NSCLC is bad, with a five-year survival rate as low as 15% [117]. Therefore, search of additional biomarkers for the development of novel treatments is mandatory. In mice, the bronchioalveolar duct junction of the adult mouse lung was suggested to harbor a stem cell population, which can be activated by K-Ras and gives rise to adenocarcinomas [118,119]. Next to bronchoalveolar stem cells, alveolar type II progenitor cells, which are Clara cells in the terminal bronchioles, were reported to directly contribute to K-Ras-induced lung hyperplasia [120,121]. K-Ras-induced tumorigenesis via type II progenitor cells in the mice lung was further shown to depend on Sox2 expression in turn affecting Notch signaling [122].

NF-κB signaling is involved in multiple steps of carcinogenesis of lung cells and mediates resistance of lung cancer cells to radio- and chemotherapy [123]. In addition, NF-κB provides one important linkage between the pathogenesis of pulmonary inflammation and cancer [124]. In this line, a recent meta-analysis [125] showed that NF-κB-positive lung cancer cells (analyzed by immunohistochemistry) are highly significantly correlated with shorter overall survival time. This might be true for both canonical and non-canonical NF-κB signaling pathways [126,127]. Not much information about NF-κB in lung CSCs is available. Only recently, CSCs were generated from the lung cancer cell lines A547 and H2170. In this model, stemness markers such as tumor sphere growth, OCT4, SOX2 and NANOG gene expression, and EMT markers were reduced after pharmacological NF-κB inhibition [128].

## 7. Cancer Stem Cells in Colon Cancer

With 450,000 new cases of colorectal cancer (CRC) in 2008 and 232,000 deaths, CRC represents the second most important cause of cancer death in Europe according to the WHO [129]. Environmental, gender, and genetic factors as well as inflammatory bowel disease like ulcerative colitis and specific prevalence of fusobacterium and other bacterial species in the microbiome of the gut are risk factors for CRC [130,131,132,133]. The aim of preoperative neoadjuvant chemo-radiation therapy is to reach downstaging/downsizing of the disease followed by surgery with improved survival [134]. Surgery is the only curative treatment modality for localized CRC. The onco-surgical approach consists of resection of the colon or rectum bearing the carcinoma, en-bloc with the draining lymph node stations. Radical lymphadenectomies, like total mesorectal excision in rectal and complete mesocolic excision in right colon cancer, have been demonstrated to improve local recurrence and overall survival [135,136]. Latent micro-metastases present at the time of surgery are thought to account for recurrence following curative colon resection. To increase the cure rate, the goal of postoperative (adjuvant) therapy is to eradicate these micro-metastases. The benefits of adjuvant chemotherapy have been most clearly demonstrated in stage III (node-positive) disease [137]. Significant progress in antibody-mediated therapy (e.g., Cetuximab (Erbitux^®^), Bevacizumab (Avastin^®^), and Panitumumab (Vectibix^®^)) was recently made [138]. There is a lot of information about normal tissue stem cells in the intestine and CRC. Normal colon stem cells were reported to be Lrg5 positive and can built cryptvillus structures in vivo [139]. But colon spheroid culture, which can initiate formation of xenografted colon carcinomas, was heterogenous in the expression Lrg5 [140]. Interestingly, single clones of CD133+/CD24+ could perform multilineage differentiation and could form tumors after xenotransplantation, albeit at a frequency of about 20%. Noteworthy, in CRC CD24, low expression is correlated with lower survival, whereas the opposite is true for breast cancer (the human protein atlas). In a mouse model of regulated expression of stable beta catenin TNF-alpha-mediated NF-kB (RelA), activation was initiated in intestinal crypt stem cells. This resulted in complete death of mice within 25 days after NF-κB activation. This was correlated with a dedifferentiation of post-mitotic intestinal epithelial cells to tumor initiating stem cells. Surprisingly, high expression of beta catenin seems to be favorable in CRC survival (the human protein atlas) [141]. NF-κB-mediated EMT as promoter of motility and invasion of epithelial cancer cells is also discussed to enable mesenchymal (stem cell) cell types out of the differentiated epithelium [142]. The axis of TGFβ/Snail with TNFα/NFκB pathways may facilitate the tumor–stroma interaction during the EMT process in CRC worsening the prognosis [143]. NF-κB-mediated EMT-inducer SNAIL1 was demonstrated to be overexpressed already in precursor lesions of CRC [144]. NF-κB-mediated EMT is negatively regulated by the scaffold protein DAB2IP. High expression of DAB2IP positively correlated with five-year survival of CRC patients. Low expression of DAB2IP is correlated with high CD133 expression and the formation or metastasis. Knockout of DAB2IP resulted in a profound increase in nuclear RelA and CD44 [145]. Further, NF-kB demonstrated the capacity to modulate tumor development in the scenario of inflammation. It was shown that inhibition of NF-κB in a metastatic mouse model of CRC-lung-metastases blocks LPS-induced tumor proliferation, enhances LPS-induced tumor apoptosis, and extends host survival [146]. Using a genetic model of intestinal epithelial cell (IEC)-restricted constitutive Wnt-activation, which comprises the most common event in the initiation of colon cancer, it was demonstrated that NF-κB modulates Wnt signaling and that IEC-specific ablation of RelA/p65 retards crypt stem cell expansion [141]. Pro-inflammatory PGE2 was demonstrated to induce CSC formation and expansion by activating NF-κB via EP4-PI3K and EP4-MAPK pathways in vitro [147]. Inhibitors of NF-κB reduced PGE2-induced sphere formation (an index of CSC expansion) and expansion of LS-174T and/or human primary CRC cells in that study. In addition, knockdown experiments revealed that NF-κB was required for PGE2 induction of CSCs and metastasis in mice. These data indicated that an inflammatory pathway, PGE2-NF-κB, mediates the effects of chronic inflammation on CSC expansion.

## 8. Prostate Cancer and Cancer Stem Cells

Urological malignancies represent a large proportion of all cancers worldwide, mainly prostate, renal cell, as well as bladder cancer [148]. Prostate cancer is the most common cancer among men in industrialized countries as the United States, whereas it is the second most common cause of cancer deaths (10%) in men [148]. All of these cancers are of epithelial origin. Acinar adenocarcinoma of the prostate is an invasive carcinoma consisting of neoplastic epithelial cells without basal cells. The neoplastic cells show a variety of histomorphological patterns: Glands, cords, single cells, and sheets (see Figure 3B).

The tumors that are grossly recognizable are firm, tan or white, and macroscopically discrete. The histological diagnosis is mainly based on the identification of invasive crowded small glands with prominent nucleoli. Different histological variants and different patterns should be considered when creating the diagnosis. Besides prostate cancer even bladder cancer and renal cell cancer are more frequent in men than in female with 7% and 5% of all male cancer cases [148]. The cancers mainly occur in the elderly from the age of 65 years and above. There is no screening tool for bladder as well as renal cell cancer. In contrary, prostate cancer can be detected in an early stage using prostate-specific-antigen-testing (PSA-testing) [149] and mortality is reduced when radical prostatectomy is performed as curative therapy [150]. Lineage tracing in mice revealed a luminal epithelial stem cell as origin of prostate cancer [151]. These cells express the homeobox-containing transcription factor Nkx3.1 and are castration resistant. RNA and protein of NKX3-1 is highly enriched in human prostate cancer (the human protein atlas). Human prostate tumor spheres (CSCs) express NKX3-1 and show constitutive NF-κB signalling with increased IL-6 production [152]. Small molecules inhibiting NF-κB significantly reduced tumor formation down to detection limits. Birnie and colleagues identified genes with altered expression in prostate cancer stem cell population such as NFKB1 and IL6 [153]. In this line, the NF-κB inhibitor parthenolide induced considerable amounts of apoptosis in both CD133-positive CSC and normal progenitor cells. Surprisingly, TNF alpha did not lead to significant nuclear translocation of p65 in sorted CD133-positve cells. More recently, it was shown that prostate stem cell antigen (PSCA), which is highly expressed also in urothelial cancer, can induce IL-6 production via NF-κB. Expression of PSCA and IL-6 were significantly associated with poor survival of patients with prostate cancer [154].

## 9. Cancer Stem Cells in Cancer of the Bone

Tumors and tumor-like lesions of the bone remain the rare tumors in humans (e.g., 1–2 primary malignant bone tumors per 100,000 inhabitants/year). In Germany, 427 men and 352 women fell ill in 2012, which corresponds to a proportion of 0.17/0.15% of all tumor diseases. In the age group of children and adolescents (age <20 years), however, the incidence is 5.6% and 4.8%, respectively [155]. The prognosis in most malignant cases is poor.

Besides the more frequent entities (like chondroma, osteochondroma, fibrous metaphyseal defect, solitary bone cyst), malignant tumors can be seldomly seen in daily clinical practice. Therefore, experiences in diagnostic and especially in therapy are concentrated on specialized tumor centers. However, due to the serious consequences of a misdiagnosis, basic knowledge has to be present in all orthopedic and trauma departments to direct the patient to the interdisciplinary tumor specialists [156].

The most common primary malignant bone tumor is the osteosarcoma. They appear in the metaphysis of long bones especially in children and young adults. Although the etiology of most sarcomas is not well known, some sarcomas are more likely to develop in certain patient populations, in association with certain genetic, infectious, and therapeutic factors [155]. As a potential cellular origin, a single cell located in the bone marrow is thought to give rise to a polyclonal, heterogeneous tumor mass [157].

Actually, also in osteosarcoma, the model of cancer stem cells is under discussion. Osteosarcoma CSCs have been demonstrated to be maintained by the stem cell transcription factor Sox2 via inhibition of the Hippo pathway [158]. Although several methods have been developed for identification of CSCs, none of these achieved to identify a pure CSC population or detect all CSC sub-populations. Accordingly, is more likely to enrich the sample for (specific) CSCs due to experimentally induced selection or, in some cases, environmental pressure [157]. Not much information about NF-κB in osteosarcoma CSCs is available.

The main identified markers in osteosarcoma CSCs are the increased presence of ABC transporters and elevated sphere formation ability accompanied by higher multidrug resistance (doxorubicin, methotrexate, cisplatin) [157]. Another marker is the expression of aldehyde dehydrogenase. Elevated expression of stemness genes like Nanog, Oct3/4, Stat3, and Sox2 were shown in CSCs with high activity of ALDH. These CSCs likewise revealed an increased self-renewal ability as well as a higher resistance to doxorubicin and cisplatin. The inhibition of ALDH activity using disulfiram resulted in reduced cell proliferation [159]. Several cell surface markers have been identified on osteosarcoma CDCs (CD133, CD117, Stro-1, and CD271) with influence on proliferation, sphere formation, and drug resistance [157]. In this regard, the major goal of current osteosarcoma CSC research is the identification of specific targets to effectively deplete CSCs during therapy [157].

## 10. Conclusions and Future Perspectives

In summary, we found compelling evidence for the pivotal role of NF-κB in organ-specific cancers. Despite the discovery of cancer stem cells in 1994, the role of NF-κB in tumor-initiating cancer stem cells is an emerging field. We suggest that further studies should be undertaken to carefully decipher the specific roles of NF-κB subunits in cancer stem cells. Furthermore, these studies may be coupled with state-of-the-art CRISPR/Cas9-mediated deletion/modification methods. In this line, we recently developed tools for specific genetic engineering of NF-κB in tumor cells [160,161], which may be applied for cancer stem cells in future studies. A common problem of cancer stem cells is their relative resistance to common transfection methods (unpublished observations), which might be relieved by the use of lentiviral transduction systems. For therapeutic intervention, cultured cancer stem cells might be an ideal platform to test combination therapy, such as antibodies, cell-mediated lysis, or chemotherapeutics and radiotherapy. In principal, therapeutically applicable targets may include CSC surface markers, their distinct niches and microenvironments, or signaling cascades mediating stemness or apoptosis of CSCs. While the suitability of surface markers for targeting CSCs is restricted in terms of marker specificity, the NF-κB family and their signaling pathways mediating both self-renewal and apoptosis of CScs may represent a promising target for therapeutic interventions. For instance, the plant-derived agent triptolide, which is in a clinical phase II trial for treating rheumatoid arthritis, was shown to downregulate NF-κB accompanied by reversed EMT and stemness features of pancreatic CSCs as well as inhibition of tumor growth [162]. Accordingly, the amount of ovarian cancers stem cells can also be reduced by blocking classical NF-κB signaling [108]. In this line, NF-κB target genes may also be of interest for therapeutic applications, as genes like TERT were also shown to be directly associated to CSC characteristics in GBM [73]. In addition, recently developed cellular therapies like those utilizing genetically engineered T cells may circumvent limitations of current treatment strategies [163] and be extended towards targeting CSCs in the future. Here, NF-κB c-Rel was recently shown to be crucial for the Treg immune checkpoint in cancer with inhibition of c-Rel in Tregs resulting in reduced melanoma growth and potentiated the effects of anti-PD-1 immunotherapy [164]. As recently reviewed by Zhang and colleagues, normal stem cells can also act to target CSCs, since these attract normal stem cells, resulting in reduced proliferation and metastasis [165]. As a summarizing clinical perspective, mechanisms of NF-κB-inhibition targeting both CSC populations within the tumor mass and maintenance of CSC characteristics may thus represent a promising strategy for novel treatment schemes.

## Figures and Tables

**Figure 1 cancers-11-00655-f001:**
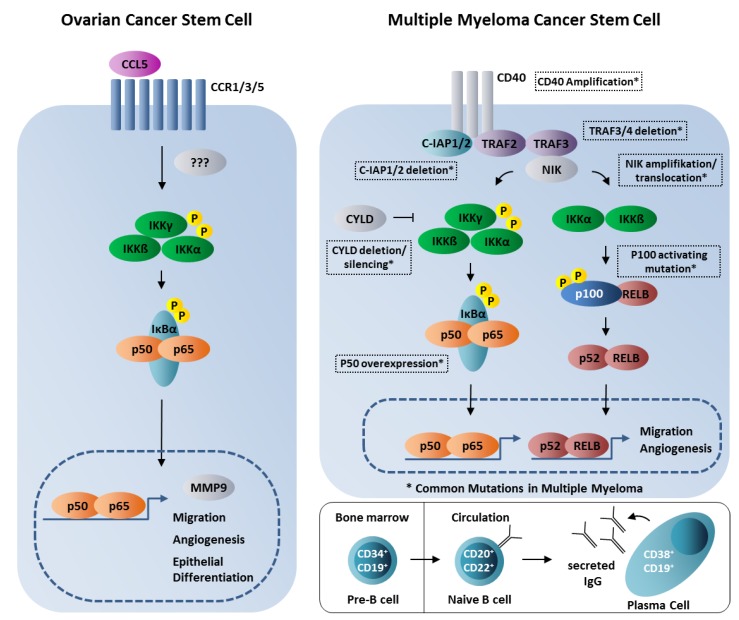
The role of NF-κB signaling pathways in cancer stem cells. Cancer stem cells (CSCs) from ovarian cancers could show CCL5-dependent activation of NF-κB p65-mediating MMP9-expression, migration, angiogenesis, and epithelial differentiation of CSCs [22] (left panel). Canonical and non-canonical NF-κB signaling in CSCs from multiple myeloma including commonly known mutations, which generate constitutive NF-κB-activity. Mutations are shown in boxes (right upper panel, modified from [23]). Development of plasma cells from pre-B cells in the bone marrow (right lower panel). Note that most authors think that NF-κB mutations are acquired during naive B cell to plasma cell differentiation, although high levels of pre-B cells in multiple myeloma patients might hint to some mutations already in present in the bone marrow.

**Figure 2 cancers-11-00655-f002:**
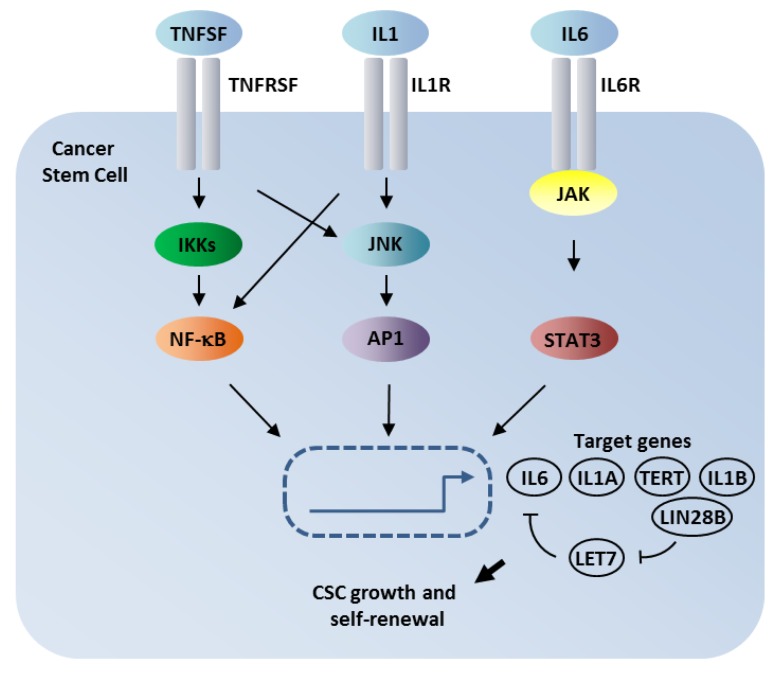
Inflammatory signaling pathways cross-coupled to NF-κB in human cancers. Two pathways (NF-κB and STAT3) were directly activated by epigenetic mechanisms in breast CSCs. Data suggest a high correlation of pro-inflammatory signaling and tumor progression. For further details, see respective text (modified from [25]).

**Figure 3 cancers-11-00655-f003:**
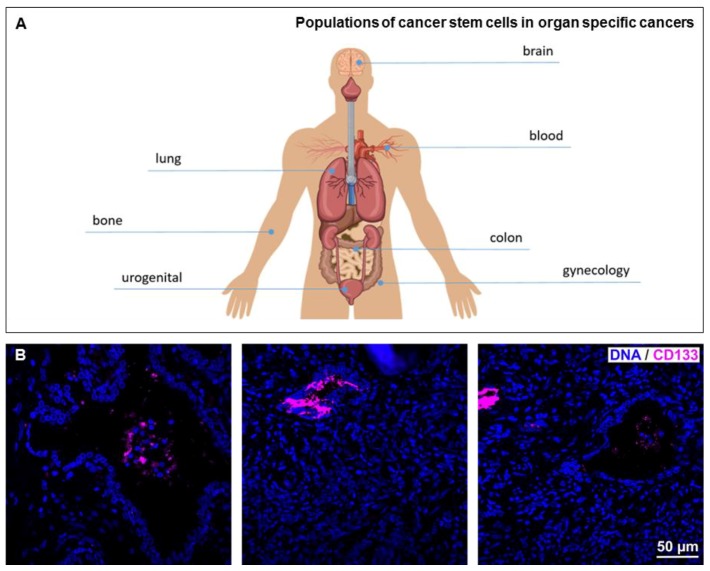
(**A**) Different organs can give rise to tumors containing cancer stem cells. (**B**) Prostate cancer tissue was identified by HE staining, stained with an antibody against the stem cell marker CD133 (magenta). Nuclei are stained with DAPI (blue). Prostate cancer typical single layer glands are visible in the left and right image. Note the nest-like appearance of CD 133 positive cancer stem cells within the cancerous tissue.

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
