# Peer review of "A Role for NF-κB in Organ Specific Cancer and Cancer Stem Cells"

_cancers, 2019, doi:10.3390/cancers11050655_

Round 1
Reviewer 1 Report
This review has cover the role of NF-kB pathway in cancer stem cells for multiple cancers. In general, the authors extensively reviewed the link between NF-kB activation and cancer initiation. However, since NF-kB pathway has dual functions in pro-inflammation and anti-inflammation, its role in cancer development may also be context dependent, particularly for inflammation related cancers. These information needs to be included. In addition, mutual cross-talks between NF-kB and many other signal pathways, including (but not limited to) Wnt and Hippo pathway in CRC, should be discussed.
Author Response
This review has cover the role of NF-kB pathway in cancer stem cells for multiple cancers. In general, the authors extensively reviewed the link between NF-kB activation and cancer initiation. However, since NF-kB pathway has dual functions in pro-inflammation and anti-inflammation, its role in cancer development may also be context dependent, particularly for inflammation related cancers. These information needs to be included. In addition, mutual cross-talks between NF-kB and many other signal pathways, including (but not limited to) Wnt and Hippo pathway in CRC, should be discussed.
We thank the referee for raising this constructive remarks. We have now discussed dual functions of NF-kB in inflammation (lines 106-114). Within this review we limited our self to the KEGG pathway “NF-kappa B signaling pathway - Homo sapiens (human)” as depicted in https://www.genome.jp/kegg-bin/show_pathway?hsa04064. Even here we simplified the canonical and non-canonical pathways for the sake of eligibility. We totally agree that Wnt and Hippo are important pathways for understanding CRC. We now included the following in our review (see lines 131-145):
Here we limited ourselves to the KEGG pathway “NF-kappa B signaling pathway - Homo sapiens (human)” as depicted in https://www.genome.jp/kegg-bin/show_pathway?hsa04064. Even here we simplified the canonical and non-canonical pathways for the sake of eligibility. It might be clear that NF-kB is cross-coupled to several other pathways. For example, there is a negative regulation of NF-kB activity by glucocorticoid receptor signaling (Auphan et al., Science. 1995 Oct 13;270(5234):286-90). Another developmentally important pathway involving secreted protein ligands of the Wnt (wingless family) shows both negative and positive action on the NF-kB pathway (as reviewed in Ma et al., Front Immunol. 2016 Sep 22;7:378.). Wnt/β-catenin pathway components could regulate responses via the interaction with the NF-κB pathway. In turn, NF-κB could influence the activity of Wnt/β-catenin signaling pathway. Specifically overexpression of beta catenin in breast and colon cancer could inhibit NF-kB activity by physical interaction, but there are reports on positive regulation (Schön et al., Int J Cancer. 2014 Oct 15;135(8):1800-11.). Furthermore, in transformed mammary epithelial cells, TGF-β and Wnt signaling synergize to induce activation of the epithelial–mesenchymal transition (EMT) program, and function in an autocrine fashion to maintain the resulting stem-cell state (Scheel et al., 2011, Cell 145: 926–940). The complex cross talk between signaling pathways of Wnt, Notch, Hippo, Hedgehog (Hh), mitogen-activated protein (MAP), kinase, phosphoinositide 3-kinase (PI3K)-Akt, and the nuclear NF-κB has been covered in a recent review (Luo, Cold Spring Harb Perspect Biol. 2017 Jan 3;9(1)).
In addition, we included a section focusing on NkappaB signaling and EMT in cancers and CSCs (see lines 115-130).

Reviewer 2 Report
Kaltschmidt et al presented a paper entitled: ‘A role for NF-kappaB in organ specific cancer and cancer stem cells. In the abstract, the authors state that the role of the transcription factor NF-kappaB in cancer stem cells is still underestimated. In fact, it is not, and there are numerous papers, if one scans the literature (pubmed).
The authors also state on pages 59-60 that Cancer stem cells………asymmetric cell division. What are the evidence that CSCs undergo asymmetric cell division?
The paper lacks a section describing the direct and indirect target genes and pathways regulated by NF-kappaB signaling that have been implicated in the generation and maintenance of CSCs and their relation to EMT (see Min et al (2008) J Cell Biochem; Qin et al (2012) Cell Res].
In Section 2. Cancer stem cells in glioblastoma multiforme, the authors describe the populations of cancer stem cells, and refer to papers that some of these CSCs express NF-kappaB which may regulate MGMT and TERT expression. But they fail to refer to experimental evidence of how NF-kappaB actually contributes to the generation and maintenance of these neuronal CSCs. For example, a recent paper suggests the operation of a TNF-NF-κB-STAT3 loop in of glioma-stem-like cells [Tafani et al (2011) J Neuroinflamm; da Hora et al (2019) Cancer Death Discov; Tannous & Badr (2019) Cell Death Disc], but also several related papers [Man et al (2014) Cell rep; Rinkenbaugh et al (2016) Oncotarget; Kim, Ezhilarasan, Phillips, et al (2016) Cancer Cell; Soubannier & Stifani (2017) Biomedicines]. The authors should have expanded on these papers. Given the title of the manuscript, the question that arises is: Are there any specific target genes and pathways affected by NF-kappaB signaling in GBM that contribute to the generation and maintenance of CSCs?
In Section 3, Cancer stem cells in pediatric cancer, the authors describe the types of pediatric cancer and the discovery of CSCs in these cancers. In the last 4-5 lines state that RelA has been detected as a fusion and that it may be involved in development. In fact, there are a number of papers on the role of NF-kappaB in development since 1995, and the authors omit several important discoveries on this issue, but this is not the issue of this paper. They provide no evidence on the role and impact of NF-kappaB signaling on pediatric cancer or on the generation and maintenance of CSCs.
In Section 4. Ovarian cancer and cancer stem cells, the authors refer to epidemiology and the contribution of BRAC genes in OC, and the OC CSC marker. They also refer to CCL5-activated NF-kappaB contributing to CSC migration, but there is no discussion on how NF-kappaB contributes to the generation and maintenance of OC CSCs. In addition, important papers are missing. For example, studies have shown a biphasic role of NF-kappaB in progression and chemoresistance of OC [Yang et al (2011) Clin Cancer Res]
In Section 5. The Role of cancer stem cells in multiple myeloma, there are many, many papers on B-cell neoplasia in mouse models related to NF-kappaB upstream activating kinases and subunits, and also on CSCs. Again there is no thorough discussion of the contribution of NF-kappaB on CSCs in this section.
In section 6. Lung cancer and cancer stem cells, the authors discuss the classification of lung cancer and epidemiology. In fact, according to cancer statistics 2018, lung cancer is the leading cause of cancer-related deaths worldwide [Bray et al (2018) CA Cancer J Clin]. Studies from mouse models provided evidence of the cell of origin of oncogene-induced NSCLC acting as a stem cell [Xu et al (2012) PNAS; Hanna & Onaitis (2013) J Carcinog; Xu et al (2014) Genes Dev], and the role and impact of NF-kappaB signaling in such mouse models [Stathopoulos et al (2007) PNAS; Meylan et al )2009) Nature; Basseres et al (2010) Cancer Res; Bivona et al (2011) Nature; Xia et al (2012) Nat cell Biol; Basseres et al (2014) Genes Cancer; Blakely et al (2015) Cell Rep], and in human lung cancer [Batra et al (2011) Arch Immunol Ther Exp; Chen et al (2011) Front Biosci]. These extremely important studies have not been taken into account and discussed in this section.
In Section 7. Cancer stem cells in colon cancer. This is an important section as it is really an example of inflammation linked to cancer, and there are numerous papers on the role and impact of NF-kappaB in colon cancer [see M. Karin’s and F. Greten’s experimental and review papers, and also Nenci et al (2007) Nature; Steinbreche et al (2008) J Immunol; Meira et al (2008) J Clin Invest; Eckmann et al (2008) PNAS; Wu et al (2009) Cancer Cell].
In Section 8. Prostate cancer and cancer stem cells, the authours discuss the epidemiology and classification of urological cancers, and briefly discuss prostate cancer (PC). They also spend a couple of lines on the constitutive activation of NF-kappaB in PC, but they should have concentrated on PC stem cells. There are several publications on this issue [Birnie et al (2008) Genome Biol; Sethi et al (2010) Am J Transl Res; Mimeault & Batra (2011) Mol Med; Rajasekhar et al (2011) Nat Comm; Deep et al (2014) Mol Cancer; Zhang et al (2016) Mol Cancer Ther[, and also in animal postate cancer models [Wang et al (2003) Cancer Cell; Wang et al (2006) Proc Natl. Acad Sci; Luo et al (2007) Nature; Mimeault & Batra (2011) BBA and references within]. There is also literature on the isolation and expansion of basal prostate progenitor cells of mouse and human origin [Hofner et al (2015) Stem Cell Rep] that the authors could refer to in the beginning of this section.
Author Response
Reviewer II
Comments and Suggestions for Authors
Kaltschmidt et al presented a paper entitled: ‘A role for NF-kappaB in organ specific cancer and cancer stem cells. In the abstract, the authors state that the role of the transcription factor NF-kappaB in cancer stem cells is still underestimated. In fact, it is not, and there are numerous papers, if one scans the literature (pubmed).
We agree with this reviewer that this specific topic has come to broader attention lately. We still believe that a comparative review on the role of NF-kB in cancer stem cells in different types of cancer is still useful and not present in this form, since we did not find another such comparative review. We therefore clarified this point in the abstract.
The authors also state on pages 59-60 that Cancer stem cells………asymmetric cell division. What are the evidence that CSCs undergo asymmetric cell division?
Thank you for raising this important question. It is reported that asymmetric cell division might be a hallmark of CSCs (Caussinus E, Hirth F. Prog Mol Subcell Biol. 2007;45:205–25). In this line a recent study using a model of colon cancer stem cells reported up to 90 % asymetric cell division as measured by life imaging (Takeda et al., Sci Rep, 2018 Dec 5;8(1):17639). We have now included these new references to better clarify this important issue (lines 62-65).
The paper lacks a section describing the direct and indirect target genes and pathways regulated by NF-kappaB signaling that have been implicated in the generation and maintenance of CSCs and their relation to EMT (see Min et al (2008) J Cell Biochem; Qin et al (2012) Cell Res].
We thank the referee for this constructive note. We have now included a section focusing on NkappaB signaling and EMT in cancers and CSCs (see lines 115-130) and also discuss cross-talks between NF-kB and the Wnt as well as Hippo pathway (see lines 131-145)
In Section 2. Cancer stem cells in glioblastoma multiforme, the authors describe the populations of cancer stem cells, and refer to papers that some of these CSCs express NF-kappaB which may regulate MGMT and TERT expression. But they fail to refer to experimental evidence of how NF-kappaB actually contributes to the generation and maintenance of these neuronal CSCs. For example, a recent paper suggests the operation of a TNF-NF-κB-STAT3 loop in of glioma-stem-like cells [Tafani et al (2011) J Neuroinflamm; da Hora et al (2019) Cancer Death Discov; Tannous & Badr (2019) Cell Death Disc], but also several related papers [Man et al (2014) Cell rep; Rinkenbaugh et al (2016) Oncotarget; Kim, Ezhilarasan, Phillips, et al (2016) Cancer Cell; Soubannier & Stifani (2017) Biomedicines]. The authors should have expanded on these papers. Given the title of the manuscript, the question that arises is: Are there any specific target genes and pathways affected by NF-kappaB signaling in GBM that contribute to the generation and maintenance of CSCs?
In the present review, we summarize evidence for TERT as a novel specific NF-kB target gene, which is associated to primary GBM an may thus play a role as a prognostic factor. TERT function was also shown to induce CSC characteristics in glioma cells via EGFR expression (Beck et al., Mol Cells. 2011 Jan;31(1):9-15), a specific NF-kB target gene often aberrant in GBM, which we already discussed as a GBM-associated marker. Pyruvate kinase M2 (PKM2), a member of the enzyme family regulating the rate-limiting step of glycolysis, is overexpressed in numerous cancers and could be regulated in glioblastoma by NF-kB (Soubannier et al., Biomedicines. 2017 Jun 9;5(2), Yang et al., Mol. Cell 2012, 48, 771–784. Angiogenesis mediated by VEGF could be also regulated by NF-kB in glioblastoma (Xie et al Oncol. Rep. 2010, 23, 725–732). Here, glioblastoma stem cells could secrete VEGF-A in extracellular vesicles (Treps et al., J Extracell Vesicles. 2017 Aug 8;6(1):1359479). Targeting the NF-kB target gene IL-6 was also demonstrated to reduce survival of CSCs in GBM concomitant with decreased tumor growth (Wang et al., Stem Cells. 2009 Oct;27:2393–404). In addition, A20 (TNFAIP3), an NF-kB target gene and regulator of cell survival, was reported to be overexpressed in CSCs from GBM with elevated levels of A20 in GSCs contributing to apoptotic resistance (Hjelmeland et al., PLoS Biol. 2010 Feb 23;8:e1000319). In line with the findings made by Rinkenbaugh an colleagues, IKK/NF-κB signaling directly contributes to glioblastoma stem cell maintenance (Rinkenbaugh et al., Oncotarget. 2016 Oct 25;7(43):69173-69187). We have now included these distinct NF-kB target genes present in CSCs from GBM within the section 2 (see lines 196-202, 211-213, 233-239).
The study by Tafani and colleaques mentioned by the referee (Tafani et al., J Neuroinflammation. 2011 Apr 13;8:32) was not included in the present review since stainings of glioblastoma cells for anti-CD133 presented in the study as FACS analysis showed nongreat difference to isotype control antibody. Therefore we were not entirely convinced that bona fide cancer stem cells were analyzed in this study. However, we now put emphasis on the TNF-NF-κB-STAT3 loop to better highlight specific NF-kB-dependent pathways in stem cells from GBM and thus shortly reviewed the following observations (see lines 219-232):
CSCs from glioblastoma seem to be highly resistant to the treatment with Smac mimetics, which are small molecule drugs inhibiting members of inhibitor of apoptosis (IAP) proteins. In CSCs from glioblastomas Smac mimetics trigger an adaptive response with increased TNFα expression and sustained activation of NF-kB and STA3 signaling (da Horta et al., Cell Death Discov. 2019 Mar 4;5:72). The cross-talk between NF-κB and STAT3 drives tumor progression and promotes stemness characteristics of cancer in multiple malignancies including gliomas (Tannous et al., Cell Death Dis. 2019 Mar 19;10(4):268.). Furthermore glioma stem cells secrete Sema3C and coordinately express Plexin A2/D1 receptors to activate Rac1/NF-κB signaling in an autocrine/paracrine loop to promote their own survival Man et al., Cell Rep. 2014 Dec 11;9(5):1812-1826). Another source of NF-κB activation in GBM involves the TGF-β/TAK1 signaling axis, which could activate both canonical and non-canonical of NF-κB signalling (Rinkenbaugh et al., Oncotarget. 2016 Oct 25;7(43):69173-69187). Proneural (PN) and mesenchymal (MES) glioma stem cells (GSCs) represent two mutually exclusive and biologically distinct GSC subtypes. Recently it was shown that mixed lineage kinase 4 (MLK4) is overexpressed in MES but not proneural PNGSCs. Silencing MLK4 suppressed self-renewal, motility, tumorigenesis, and radioresistance of MES. MLK4 bound and phosphorylated the NF-κB regulator IKKα, leading to activation of NF-κB signaling in MES (Kim et al., Cancer Cell. 2016 Feb 8;29(2):201-13).
In Section 3, Cancer stem cells in pediatric cancer, the authors describe the types of pediatric cancer and the discovery of CSCs in these cancers. In the last 4-5 lines state that RelA has been detected as a fusion and that it may be involved in development. In fact, there are a number of papers on the role of NF-kappaB in development since 1995, and the authors omit several important discoveries on this issue, but this is not the issue of this paper. They provide no evidence on the role and impact of NF-kappaB signaling on pediatric cancer or on the generation and maintenance of CSCs.
We thank the referee for raising this issue. From our point of view, we summarize evidence for the role of NF-kB RELA in fusion with C11orf95 in transformation of neural stem cells and initiation of CSC-driven tumorigenesis. This matter is now discussed in more detail to better guide the reader through this section (lines 286-287).
In Section 4. Ovarian cancer and cancer stem cells, the authors refer to epidemiology and the contribution of BRAC genes in OC, and the OC CSC marker. They also refer to CCL5-activated NF-kappaB contributing to CSC migration, but there is no discussion on how NF-kappaB contributes to the generation and maintenance of OC CSCs. In addition, important papers are missing. For example, studies have shown a biphasic role of NF-kappaB in progression and chemoresistance of OC [Yang et al (2011) Clin Cancer Res]
Concerning the contribution of NF-kB to the generation and maintenance of OC CSCs, we already discuss that blocking of classical NF-kB pathway results in reduction of CD44+ CSCs, thus linking maintenance of CSCs to NF-kB activity. The study by Yang et al. 2011 used cell lines such as the high-grade serous ovarian cancer cell lines SKOV3, HEY, SKOV3.ip1, HEYA8, OVCAR3 but did not specifically analyse CSC. Therefore this important work is not included in this review focusing on CSCs.
In Section 5. The Role of cancer stem cells in multiple myeloma, there are many, many papers on B-cell neoplasia in mouse models related to NF-kappaB upstream activating kinases and subunits, and also on CSCs. Again there is no thorough discussion of the contribution of NF-kappaB on CSCs in this section.
We totally agree there is a lot of evidence in mouse models. However, there is a big difference in innate immunity between mouse and men as reviewed (Zschaler et al., Crit Rev Immunol. 2014;34(5):433-54). To keep focus we have mainly concentrated on summarizing and discussing studies utilizing human CSCs.
In section 6. Lung cancer and cancer stem cells, the authors discuss the classification of lung cancer and epidemiology. In fact, according to cancer statistics 2018, lung cancer is the leading cause of cancer-related deaths worldwide [Bray et al (2018) CA Cancer J Clin]. Studies from mouse models provided evidence of the cell of origin of oncogene-induced NSCLC acting as a stem cell [Xu et al (2012) PNAS; Hanna & Onaitis (2013) J Carcinog; Xu et al (2014) Genes Dev], and the role and impact of NF-kappaB signaling in such mouse models [Stathopoulos et al (2007) PNAS; Meylan et al )2009) Nature; Basseres et al (2010) Cancer Res; Bivona et al (2011) Nature; Xia et al (2012) Nat cell Biol; Basseres et al (2014) Genes Cancer; Blakely et al (2015) Cell Rep], and in human lung cancer [Batra et al (2011) Arch Immunol Ther Exp; Chen et al (2011) Front Biosci]. These extremely important studies have not been taken into account and discussed in this section.
We thank the reviewer for these helpful notes. Although the latest numbers of the GLOBOCAN 2018, World Health Organization, referring that lung cancer is taking the first place in its incidence as well as in its mortality rates worldwide, a deeper analysis of their data showed that lung cancer still behind breast and prostatic cancer according to the “Age-standardized incidence rate per sex (Fig. 1).
Similar to section 5, we have also concentrated on the human system to keep focus. We have now included the respective studies by Batra et al. and Chen et al. within section 6 (see lines 376-378).
In Section 7. Cancer stem cells in colon cancer. This is an important section as it is really an example of inflammation linked to cancer, and there are numerous papers on the role and impact of NF-kappaB in colon cancer [see M. Karin’s and F. Greten’s experimental and review papers, and also Nenci et al (2007) Nature; Steinbreche et al (2008) J Immunol; Meira et al (2008) J Clin Invest; Eckmann et al (2008) PNAS; Wu et al (2009) Cancer Cell].
We are very well familiar with Greten´s work with Michael Karin, but this all involves mouse and cancer, not cancer stem cells. Greten´s recent work on CSCs in colon cancers is discussed within the present review (Schwitallla et al., 2013). Similarly, Pasparaki´s work on Nemo in mice (Nenci et al. 2007) does no cover human CSCs and is therefore not included within the present manuscript. However, we now included a more detailed discussion regarding the role of NF-kB-signaling in CSCs in the context of inflammation (lines 424-436).
In Section 8. Prostate cancer and cancer stem cells, the authours discuss the epidemiology and classification of urological cancers, and briefly discuss prostate cancer (PC). They also spend a couple of lines on the constitutive activation of NF-kappaB in PC, but they should have concentrated on PC stem cells. There are several publications on this issue [Birnie et al (2008) Genome Biol; Sethi et al (2010) Am J Transl Res; Mimeault & Batra (2011) Mol Med; Rajasekhar et al (2011) Nat Comm; Deep et al (2014) Mol Cancer; Zhang et al (2016) Mol Cancer Ther[, and also in animal postate cancer models [Wang et al (2003) Cancer Cell; Wang et al (2006) Proc Natl. Acad Sci; Luo et al (2007) Nature; Mimeault & Batra (2011) BBA and references within]. There is also literature on the isolation and expansion of basal prostate progenitor cells of mouse and human origin [Hofner et al (2015) Stem Cell Rep] that the authors could refer to in the beginning of this section.
We thank the reviewer for these improving remarks. We now included the following observation into the present review (see lines 459-463): Birnie and colleagues identified genes with altered expression in prostate cancer stem cell population such as NFKB1 and IL6 (Birnie et al., Genome Biol. 2008;9(5):R83). In this line the NF-kB inhibitor parthenolide induced considerable amounts of apoptosis in both CD133-positive CSC and normal progenitor cells. Surprisingly, TNF alpha did not lead to significant nuclear translocation of p65 in sorted CD133-positve cells.
Since we focus on cancer-related stem cells, the study by Hofner et al. was not included into the manuscript, while Deep et al. 2014 were analyzing the PC3 prostate carcinoma cell line and therefore not included into the present review.

Reviewer 3 Report
The authors described the involvement of NF-kappaB (NF-kB) in multiple cancers and major focus on cancer stem cells (CSCs), a important subpopulation of cancer cells participating in tumor initiation, drug resistance and metastasis. The provided information are a lot and the figure 2 is informative, but the arrangement of manuscript, which was according to types of cancer, was can be improved. I think such arrangement is not good for emphasize the importance of NF-kB in the maintenance of CSCs. The better way of such review is to describe the functions/populations of CSCs in different types of cancers at first. Then to provide the evidences of the involvement of NF-kB in CSCs including the different ways of NF-kB activation among CSC types (for example, by inflammation or growth factor receptor signal) and the behavior of CSCs regulated by NF-kB among cancer types (for example, to divide into cancer initiation, resistance to treatment, or invasive phenotype). Finally, I will encourage authors to provide a section of "Future perspective" to provide the future direction for NF-kB based studies or drug development in cancer biology. Such kind of manuscript arrangement will provide a better thread for readers to get a whole picture for the involvement of NF-kB in CSCs.
Author Response
The authors described the involvement of NF-kappaB (NF-kB) in multiple cancers and major focus on cancer stem cells (CSCs), a important subpopulation of cancer cells participating in tumor initiation, drug resistance and metastasis. The provided information are a lot and the figure 2 is informative, but the arrangement of manuscript, which was according to types of cancer, was can be improved. I think such arrangement is not good for emphasize the importance of NF-kB in the maintenance of CSCs. The better way of such review is to describe the functions/populations of CSCs in different types of cancers at first. Then to provide the evidences of the involvement of NF-kB in CSCs including the different ways of NF-kB activation among CSC types (for example, by inflammation or growth factor receptor signal) and the behavior of CSCs regulated by NF-kB among cancer types (for example, to divide into cancer initiation, resistance to treatment, or invasive phenotype). Finally, I will encourage authors to provide a section of "Future perspective" to provide the future direction for NF-kB based studies or drug development in cancer biology. Such kind of manuscript arrangement will provide a better thread for readers to get a whole picture for the involvement of NF-kB in CSCs.
We thank the referee for rasing these helpful remarks. We now changed the “conclusion” section towards “conclusion and future perspective” and provided additional future directions emphasizing the role of NF-kB in the biology of cancer biology (see lines 529-546).
To emphasize the importance of NF-kB in the maintenance of CSCs, we have now included new paragraphs into the introduction section. We now also discuss the role of NF-kB as a mediator of inflammation in the context of cancer progression and emphasize crosslinks between NF-kB signaling and other crucial cancer-driving mechanisms like EMT, Wnt/β-catenin or the Ras-/TGF-ß-pathway in cancer and cancer stem cells (see lines 115-145). Afterwards, we still focus on distinct CSC populations and their functions in cancer progression in the context of the respective organ. As in the originally submitted manuscript, we provide evidences of the involvement of NF-kB in CSC biology and function. However, from our point of view a context (organ)-dependent summary and discussion of NF-kB signaling in CSCs seems more suitable for assuring readability and guidance of the reader through this highly complex field. In this regard, we initially included Figure 1 as a first overview on the range of populations of CSCs discussed in the present review. To conclude our discussions regarding the role of NF-kB signaling pathways in cancer stem cells, we including figure 2 to provide a graphical overview and extended our conclusion section towards future perspectives as mentioned above.

Round 2
Reviewer 2 Report
Kaltschmidt et al resubmitted athe paper entitled: ‘A role for NF-kappaB in organ specific cancer and cancer stem cells, following revision. However, there are still some issues to be discussed.
The authors added (lines 63-66): Interestingly, especially asymmetric cell division (ACD) is discussed for being a hallmark of CSCs [5]. See also Mukherjee et al (2015) Stem Cells and Dev 24:405-16. Since the paper presented is on NF-kappaB in CSCs, what are the evidence for the implication of NF-kappaB signaling in ACD of CSCs? It has also been reported that transient oncogene induction in non-transformed, immortalised human cells can result in elevated IL-6 production, which maintains cells in a transformed self-renewing state containing cancer stem cells by driving an epigenetic, positive feedback loop to simultaneously activate NF-κB and STAT3 [Iliopoulos D et al (2009) Cell 139:693-706]. In keeping with this, in Biomedicines (another MDPI journal), a paper on the NF-kappaB – miRNA regulatory network in cancer, also discussing CSCs was published recently, and should be taken into account [Markopoulos G et al (2018) Biomedicines 6(2):40, and also Markopoulos et al (2017) Cell Oncol 40:303-39], also promoting MDPI journals.
From lines 76 to 93, the authors give a brief description of NF-kappaB signaling, but omit to n discuss the role of IKK-mediated NF-kappaB-dependent or independent signaling. The IKK kinases may also be implicated in CSC biology. I quite like to see a figure on NF-kappaB signaling and how it impacts on CSCs.
On line 95, the authors state: In particular, NF-kappaB mediates vital tumor-promoting mechanisms…..Do they refer to canonical or noncanonical signaling? They should be specific.
In Section 4. Ovarian cancer and cancer stem cells, the authors refer to the contribution of BRAC genes in OC, and the OC CSC marker. The authors should refer to the connection of BRAC and NFkappaB in OC [Volcic et al (2012) Nucleic Acids Research, 40:181-195; Harte et al (2014) Oncogene 33:713-23]..
In Section 5. The Role of cancer stem cells in multiple myeloma, there are many, many papers on B-cell neoplasia, which the authors should include in their paper [For example: Sasaki et al (2006) Immunity 24:729-39; Hunter et al (2016) Oncogene 30:3476-84; Hunter et al (2016) Br J Cancer 12:1-6; Derudder et al (2016) PNAS 113:5065-70]
In section 6. Lung cancer and cancer stem cells, I made several comments, but the authors only added the paper by Batra et al (2011), but they have not discussed the other papers discussing the emergence and importance of cancer stem cells in NSCLC, but also the role of NF-kappaB signaling in lung CSCs [Berns A (2005) Cell; Kim CF et al (2005) Cell 121:823-35; Xu et al (2012) PNAS; Hanna & Onaitis (2013) J Carcinog; Xu et al (2014) Genes Dev]. The authors should discuss and include these important papers. Although these are mostly on mouse lung cancer models but are equally important for human cancer.
I hope that the authors will take into account my comments. I know that it is frustrating to look over and over again a submitted manuscript, but often comments improve the quality of manuscripts and the impact of journals.
Author Response
Response to Reviewer II
Kaltschmidt et al resubmitted the paper entitled: ‘A role for NF-kappaB in organ specific cancer and cancer stem cells, following revision. However, there are still some issues to be discussed.
We very much appreciate the time and effort of the referee to help us in improving our review. We have addressed all suggestions as detailed below.
The authors added (lines 63-66): Interestingly, especially asymmetric cell division (ACD) is discussed for being a hallmark of CSCs [5]. See also Mukherjee et al (2015) Stem Cells and Dev 24:405-16.
We now added Mukkerjee et al. as new reference (see line 63).
Since the paper presented is on NF-kappaB in CSCs, what are the evidence for the implication of NF-kappaB signaling in ACD of CSCs?
We thank the referee for raising this improving question. To the best of our knowledge, not many publications are available regarding symmetric cell division and NF-kB in CSCs. For example, breast cancer stem cells express the receptor NOTCH, which could be stimulated by NF-kB-mediated expression of its ligand JAG1 on non-cancer stem cells. Thus, proliferation of CSCs can be triggered in trans by an NF-kB-dependent mechanism (Yamamoto et al., Nat Commun. 2013; 4:2299.). In this line, the NOTCH pathway is thought to be one of the crucial survival pathways of CSCs (Borah et al., Oncogenesis. 2015 Nov 30;4:e177. doi: 10.1038/oncsis.2015.35.), although activation of NOTCH pathway was also described to activate NF-kB in various non-tumor cells (see Osipo et al., Lab Invest. 2008 Jan;88(1):11-7) for review). NOTCH is famous for its role in regulating a switch from symmetric to asymmetric cell divison in neural stem cells of Drosophila, but the role of NOTCH as a regulator of NF-kB in CSCs has not yet been investigated to that rigor, which is possible when using Drosophila genetics. We now included these aspects within the revised manuscript (see below and lines 65-67).
Lines 65-67: “In this line, breast cancer stem cells express the receptor Notch, which could be stimulated by NF-kB-mediated expression of its ligand JAG1 on non-cancer stem cells. Thus, proliferation of CSCs can be triggered by an NF-kB-dependent mechanism [7].”
It has also been reported that transient oncogene induction in non-transformed, immortalized human cells can result in elevated IL-6 production, which maintains cells in a transformed self-renewing state containing cancer stem cells by driving an epigenetic, positive feedback loop to simultaneously activate NF-κB and STAT3 [Iliopoulos D et al (2009) Cell 139:693-706].
Thank you for providing this reference, we now included this information also in the novel figure describing the impact of NF-kB on cancer stem cells (see revised Figure 2, Lines 138-139). Using mammospheres induced by overexpression of the SRC oncogene as a model of CSCs development in breast cancer, an epigenetic switch was discovered by Iliopoulos and coworkers (Cell. 2009 Nov 13;139(4):693-706). In particular, a short period (24h) of induction by SRC was sufficient to change gene expression in a manner that NF-kB and STAT3 signaling became dominant (see revised Figure 2). Furthermore, expression of transcription factor LIN28 was induced, which is a repressor of LET7 microRNA, thus releasing the repression of IL6 protein expression. Recent findings elaborated these observations to a more general model for the regulation of an inflammatory regulatory network in many human cancers (see revised Figure 2). This network involves three major inflammatory pathways: NF-kB, which can be activated by members of the TNF superfamily (TNFSF) and IL1, the transcription factor AP1 (JUN/FOS) and STAT3. All three transcription factors are activated by transmembrane receptors, which impinge on kinases (IKKs for NF-KB, JNK for AP1 and JAK for STAT3; Ji et al., Proc Natl Acad Sci U S A. 2019 Mar 25. pii: 201821068). We now included these discussion in the revised manuscript (see revised Fig. 2 and below as well as lines 125-137).
Lines 125-137: “With regards to maintainance of self-renewal of CSCs, Iliopoulos and coworkers discovered an epigenetic switch in mammospheres induced by overexpression of the SRC oncogene as a model of CSCs development in breast cancer [22]. In particular, a short period (24h) of induction by SRC was sufficient to change gene expression in a manner that NF-kB and STAT3 signaling became dominant (see Fig. 2). Furthermore, expression of transcription factor LIN28 was induced, which is a repressor of LET7 microRNA, thus releasing the repression of IL6 protein expression. Recent findings elaborated these observations to a more general model for the regulation of an inflammatory regulatory network in many human cancers (see Fig. 2). This network involves three major inflammatory pathways: NF-kB, which can be activated by members of the TNF superfamily (TNFSF) and IL1, the transcription factor AP1 (JUN/FOS) and STAT3. All three transcription factors are activated by transmembrane receptors, which impinge on kinases (IKKs for NF-kB, JNK for AP1 and JAK for STAT3, see Fig. 2 and [23]).”
In keeping with this, in Biomedicines (another MDPI journal), a paper on the NF-kappaB – miRNA regulatory network in cancer, also discussing CSCs was published recently, and should be taken into account [Markopoulos G et al (2018) Biomedicines 6(2):40, and also Markopoulos et al (2017) Cell Oncol 40:303-39], also promoting MDPI journals.
The balance of activating and repressing miRNA on transcription factor signaling and cancer is reviewed by Lüningschrör et al., Biochim Biophys Acta. 2013; 1833(8):1894-903 and Markopoulos et al.,
Biomedicines. 2018 Mar 30;6(2)., also promoting MDPI journals. Clearly, miRNAs are important regulators of NF-kB. We could previously show that NF-kB RelA protein is repressed by microRNAs in embryonic stem cells and that this repression is released during neuronal differentiation (Lüningschrör et al., Stem Cells. 2012 Apr;30(4):655-64). Markopoulos and coworkers reviewed more than 150 miRNAs acting on NF-kB in cancer. A well-established feed-back loop is the repression of miRNA LET7 by embryonic transcription factor LIN28B in cancer stem cells (see revised Fig. 2). These aspects are now included within the revised manuscript (see below and lines 145-149).
Lines 145-149: “In line with the depicted feed-back loop of the repression of miRNA LET7 by embryonic transcription factor LIN28B in cancer stem cells (Fig. 2), Markopoulos and coworkers recently reviewed more than 150 miRNAs acting on NF-kB in cancer [24]. Accordingly, we could previously show that NF-kB RelA protein is repressed by microRNAs in embryonic stem cells and that this repression is released during neuronal differentiation [25,26].”
From lines 76 to 93, the authors give a brief description of NF-kappaB signaling, but omit to n discuss the role of IKK-mediated NF-kappaB-dependent or independent signaling. The IKK kinases may also be implicated in CSC biology. I quite like to see a figure on NF-kappaB signaling and how it impacts on CSCs.
We thank the referee for raising this issue. We now included a novel figure (revised figure 2) in the revised manuscript, which depicts novel insults on the interplay of inflammatory pathways, including the derepressing action of transciption factor LIN28 on IL6 protein expression. We also included the role of IKK in mediating NF-kB activation upon stimulation of CSCs with TNFSF or IL1. In addition, distinct roles of IKKs in ovarian cancer stem cells as well as multiple myeloma stem cells are depicted in figure 1.
On line 95, the authors state: In particular, NF-kappaB mediates vital tumor-promoting mechanisms…..Do they refer to canonical or noncanonical signaling? They should be specific.
To the best of our knowledge, most data point to a role of canonical NF-kB pathway in mediating functionality of CSCs, together with AP1 and STAT3 (see revised figure 2 and lines 145-149). We now changed the respective statement accordingly: “In particular, canonical NF-kB signaling mediates vital tumor-promoting mechanisms […]” (see lines 97-98).
In Section 4. Ovarian cancer and cancer stem cells, the authors refer to the contribution of BRAC genes in OC, and the OC CSC marker. The authors should refer to the connection of BRAC and NFkappaB in OC [Volcic et al (2012) Nucleic Acids Research, 40:181-195; Harte et al (2014) Oncogene 33:713-23]..
Thank you for suggesting this interesting aspect, which we now included together with the respective references within the revised manuscript (see below and lines 322-328).
Lines 322-328: “Notably, BRCA1 was shown to associate constitutively with NF-κB p65, while p50 associated only with BRCA1 upon DNA damage treatment. Harte and coworkers thus described a functional interdependence between BRCA1 and NF-kB, which is required for BRCA1-mediated resistance to DNA damage [92]. In addition, NF-κB was demonstrated to directly regulate DNA double-strand break repair in a BRCA2-dependent manner as well as via interaction with CtIP-BRCA1 complexes and promotion of BRCA1 stabilization [93].”
In Section 5. The Role of cancer stem cells in multiple myeloma, there are many, many papers on B-cell neoplasia, which the authors should include in their paper [For example: Sasaki et al (2006) Immunity 24:729-39; Hunter et al (2016) Oncogene 30:3476-84; Hunter et al (2016) Br J Cancer 12:1-6; Derudder et al (2016) PNAS 113:5065-70]
We now included the respective references within the revised manuscript (see below and lines 381-385).
Lines 381-385: “We plan to analyze the contribution of NF-kB to MMSC. In this line most NF-kB activating mutations were attributed to mutations acquired by a stromal independent multiple myeloma cell (see [110] for discussion). Notably, canonical NF-kB signaling was already observed to be required for persistence as well as functionality of mature B cells [113] and is a hallmark of various B cell lymphomas [114]. NF-kB cRel is further known as a key regulator of B-cell proliferation and is frequently amplified in B-cell lymphoma [115], but was also described to have a tumor suppressor role in lymphoma development [116]. In figure 1, we summarize hypotheses for the generation of multiple myeloma and the activation of NF-kB.”
In section 6. Lung cancer and cancer stem cells, I made several comments, but the authors only added the paper by Batra et al (2011), but they have not discussed the other papers discussing the emergence and importance of cancer stem cells in NSCLC, but also the role of NF-kappaB signaling in lung CSCs [Berns A (2005) Cell; Kim CF et al (2005) Cell 121:823-35; Xu et al (2012) PNAS; Hanna & Onaitis (2013) J Carcinog; Xu et al (2014) Genes Dev]. The authors should discuss and include these important papers. Although these are mostly on mouse lung cancer models but are equally important for human cancer.
We now included the respective references within the revised manuscript (see below and lines 400-406).
Lines 400-406: In mice, the bronchioalveolar duct junction of the adult mouse lung was suggested to harbor a stem cell population, which can be activated by K-Ras and gives rise to adenocarcinomas [118,119]. Next to bronchoalveolar stem cells, alveolar type II progenitor cells, which are Clara cells in the terminal bronchioles, were reported to directly contribute to K-Ras-induced lung hyperplasia [120][121]. K-Ras-induced tumorigenesis via type II progenitor cells in the mice lung was further shown to depend on Sox2 expression in turn affecting Notch signaling [122].
I hope that the authors will take into account my comments. I know that it is frustrating to look over and over again a submitted manuscript, but often comments improve the quality of manuscripts and the impact of journals.

Reviewer 3 Report
The authors have greatly improved their manuscript and provided informative knowledge in the role of NF-kB in different types of cancer stem cells. The manuscript could be accepted for publication.
Author Response
The authors have greatly improved their manuscript and provided informative knowledge in the role of NF-kB in different types of cancer stem cells. The manuscript could be accepted for publication.
We thank the referee very much for suggesting our manuscript to be acceptable for publication.
Round 3
Reviewer 2 Report
I am now satisfied with the answers of the authors to my comments, and the additions to the text as I suggested.